# Damage Quantification with Embedded Piezoelectric Aggregates Based on Wavelet Packet Energy Analysis

**DOI:** 10.3390/s19020425

**Published:** 2019-01-21

**Authors:** Zijian Wang, Li Wei, Maosen Cao

**Affiliations:** 1State Key Laboratory of Hydrology-Water Resources and Hydraulic Engineering, Nanjing Hydraulic Research Institute, Nanjing 210029, China; zijianwang@nhri.cn; 2School of Mechanics and Materials, Hohai University, Nanjing 210024, China

**Keywords:** smart aggregate, piezoelectric material, structural health monitoring, elastic waves, damage quantification, wavelet packet analysis, cement material, finite element modeling, explicit analysis

## Abstract

Cement-based components have been widely used in civil engineering structures. However, due to wearing and deterioration, the cement-based components may have brittle failure. To provide early warning and to support predictive reinforcement, the piezoelectric materials are embedded into the cement-based components to excite and receive elastic waves. By recognizing the abnormalities in the elastic waves, hidden damage can be identified in advance. However, few research has been published regarding the damage quantification. In this paper, the wavelet packet analysis is adopted to calculate the energy of the transmitted elastic waves based on the improved piezoelectric aggregates (IPAs). Due to the growth of the damage, less elastic waves can pass through the damage zone, decreasing the energy of the acquired signals. A set of cement beams with different crack depths at the mid-span is tested in both numerical and experimental ways. A damage quantification index, namely the wavelet packet-based energy index (WPEI), is developed. Both the numerical and experimental results demonstrate that the WPEI decreases with respect to the crack depth. Based on the regression analysis, a strong linear relationship has been observed between the WPEI and the crack depth. By referring to the linear relationship, the crack depth can be estimated by the WPEI with a good accuracy. The results demonstrated that the use of the IPAs and the WPEI can fulfill the real-time quantification of the crack depth in the cement beams.

## 1. Introduction

Abrupt structural failures may cause devastating casualties. A safe and economical way to prevent this kind of tragedy is to identify the damage as early as possible and thus to perform necessary reinforcements in advance. To this end, significant efforts have been made over the past few decades towards the development of the structural health monitoring system. Based on the network that consists of a large number of embedded and/or attached sensors [1], the structural health monitoring system can reflect the structural operation state in real time by monitoring the variations of various physical parameters, such as the modal parameter [2], the electromechanical admittance or impedance [3], and the acoustic wave [4]. Among these damage-related physical parameters, the acoustic parameters [5] have been demonstrated as nondestructive, sensitive, and robust indicators for the damage identification. By analyzing the abnormal reflection and attenuation of the elastic waves, hidden damage can be detected, located, and quantified by the sensor network.

The implementation of the acoustic-based damage identification mainly depends on the piezoelectric ceramics, which can convert the mechanical stress to the electrical voltage, and vice versa [6]. Based on this advantage, the piezoelectric ceramics can operate as both the actuator and the receiver to form the sensor network to monitor a substantial area. However, the piezoelectric ceramics are brittle for direct embedment in the cement-based engineering structures. To fix this problem, the concept of the piezoelectric aggregate (smart aggregate) has been proposed [7]. By sealing the ceramic patch with waterproof adhesives and cement blocks, the piezoelectric aggregate can strengthen the ceramic patch to sustain the potential impact during the concrete casting process. With this configuration, the piezoelectric aggregate can be placed deeply into the structure to form a global network and collect the data from inaccessible locations. By comparing the elastic wave signals with the baselines, the damage can be detected based on the pitch-catch fashion of two piezoelectric aggregates [8].

Kong et al. [9,10] developed a spherical piezoelectric aggregate by radially polarizing the ceramic shell. With this spherical piezoelectric aggregate, the elastic waves can be equally excited and received in all directions. Wang et al. [11] developed a theoretical model to depict the vibration characteristics of the spherical piezoelectric aggregate. A dynamic analytical solution of the aggregate subjected to a harmonic voltage excitation was derived along with an analytical expression of the electrical impedance. Yan et al. [12] developed a wireless piezoelectric aggregate to acquire the structural response due to the dynamic stress, impact capturing, and internal crack. Song et al. [13] used the composite materials to pack a piezoelectric ceramic chip and employed the 3-D printing technique to build a high-quality mold for aggregate casting. Dumoulin and Deraemaeker [14] developed a simplified finite element model with the multi-objective genetic algorithm to explore the working principles of the embedded piezoelectric patches in the infinite medium. Zou et al. [15] employed the electronic universal testing machine to investigate the performance of the piezoelectric aggregate under various compressive stresses. Tian el al. [16] adopted the Rayleigh damping model to characterize the elastic wave attenuation in the concrete. The model indicates that the absorption attenuation coefficient is proportional to the square of the wave frequency when the damping ratio is small. The results of an experiment based on the time-reversal method accord with the theoretical model. Lim et al. [17] conducted a series of semi-analytical and numerical investigations into the elastic wave propagation during the concrete curing process. A set of parameters including the elastic modulus, the Poisson’s ratio, and the damping coefficient were discussed. Ai et al. [18] developed a 2-D analytical model to depict the interaction between the embedded piezoelectric transducer and the host structure. This model is successively validated by an experiment on a reinforced concrete beam.

However, since the cement-based materials usually contain randomly distributed gravels and voids, the propagations of the elastic waves are seriously distorted and attenuated due to the multiple refractions at the interfaces of discontinuity. To interpret the complicated wave signals and extract the damage-related wave features, the acquired signals need to be non-linearly transformed to form damage indices which can characterize the damage location and severity. Some widely reported damage indices are formulated based on the Fourier, the Hilbert–Huang, and the wavelet (packet) transformations.

Gao et al. [19] utilized the time difference between the wave arrivals as a damage index and employed the delay-and-sum imaging method to locate the damage in the 2-D concrete structure through colorful contours. Zhang et al. [20] utilized the wavelet packet analysis to develop a damage index to detect the damage in the L-shaped concrete-filled steel tube. Yan et al. [21] utilized the energy of elastic waves as a damage index to monitor the debonding defect between the steel and concrete. Markovic et al. [22] proposed a damage index based on the root-mean-square deviation to fulfill the real-time monitoring of the concrete cracking. Xu et al. [23] investigated the influence of the concrete meso-scale structure on the response of the embedded piezoelectric aggregate. The interface debonding in the concrete-filled steel tube is detected by a damage index based on the wavelet packet analysis. Chen et al. [24] utilized the wavelet packet analysis to develop a damage index to detect the concrete aggregate segregation based on the multi-scale and multi-physical field coupling simulation. Feng et al. [25] conducted the wavelet packet-based energy analysis to characterize the energy of the elastic waves. The experimental results show that the elastic wave energy has the potential to characterize the crack developing and repairing process in the concrete structures.

With the multifunctional piezoelectric aggregates and the advanced signal processing methods, various types of damage have been successfully detected with high sensitivity and reliability.

Wu et al. [26] investigated the feasibility of the embedded piezoelectric aggregates to detect the interlayer landslide. The results indicate that the occurrence of the interlayer slide attenuates the wave energy and decreases the signal intensity. Hou et al. [27] utilized the piezoelectric aggregates to monitor the compactness of the compound concrete filled with the demolished concrete lumps. Zhang et al. [28] predicted the seismic stress by analyzing the average output voltages of a limited number of embedded piezoelectric aggregates in the high-strength concrete. Markovic et al. [29] proposed a hybrid approach using both the wave propagation energy and the time-of-flight to locate the damage in the concrete plate based on the finite element simulation. Liu et al. [30] investigated the axial compressive loading on the monitoring results of the piezoelectric aggregates. Three loading cases, including the single cycle, the cyclic, and the step-by-step load, were tested experimentally. The results show that the axial loads, less than 30% of the failure load, have a significant impact on the elastic wave signals. The amplitude attenuation is dependent on both frequency and load history, while the velocity is highly dependent on stress. Dumoulin and Deraemaeker [31] proposed a new data acquisition system with a high sampling frequency, a low power consumption, and a high signal-to-noise ratio. Therefore, the system is able to capture the brittle failure events during the concrete compression process. Karayannis et al. [32] detected the damage stages of a simply supported beam under flexural loadings. By investigating the electromechanical admittance of the embedded piezoelectric sensors, the concrete cracking and the steel bar yielding have been successfully detected. Voutetaki et al. [33] developed a portable circuit board to wirelessly transmit the monitoring signal. Based on the electromechanical impedance, the crack due to the shear loading is detected for a reinforced concrete beam.

On the other hand, a few studies have been recently conducted to investigate the relationship between the elastic wave energy and the damage severity.

Jiang et al. [34] conducted a set of experiments trying to provide the early warning of the initial corrosion for the prestressed concrete structures. The experimental results show that the elastic wave energy slightly increases in the initial corrosion stage due to the internal expansion pressure caused by the corrosion products. Subsequently, with the occurrence and the development of the corrosion-induced cracks, the elastic wave energy decreases as the corrosion develops. When the corrosion process almost completes, the elastic wave energy becomes stable. Du et al. [35] presented an experimental study by setting up the piezoelectric transducers to investigate the relationship between the porosity in cement paste and the elastic wave parameters. The actual pore structures were measured by the X-ray computed tomography. The results indicate that the signal amplitude and velocity decrease as the material porosity increases. Xu et al. [36] adopted the wave attenuation ratio to detect the presence of the low strength concrete in the beam-column joint. The experimental results show that the lower strength of the concrete causes increasing wave attenuation.

Although the abovementioned studies have demonstrated that the elastic wave energy has a strong correlation with the damage severity, not much research has been reported regarding the damage quantification, which is at a more sophisticated level. To this end, a damage quantification index is proposed in this paper to quantify the crack depth in the cement beams. By carefully designing the damage index based on the wavelet packet analysis, a linear relationship is obtained between the index and the crack depth. Furthermore, an improved piezoelectric aggregate (IPA) is developed based on a new method of preparation. The IPA is able to effectively alleviate the cross talk in the acquired signal and thus to facilitate the interpretation of the wave signals.

## 2. Method of Damage Quantification

### 2.1. Elastic Waves in Cement Beams

The deformation of the piezoelectric aggregates can generate two kinds of elastic waves in the cement beams: the longitudinal and the transverse waves. By characterizing the cement materials as infinite isotropic media, the propagation velocity of the longitudinal and the transverse waves can be described as
(1)$$\begin{array}{l} C_{L}=\sqrt{\frac{E\left(1-\mu \right)}{\rho \left(1+\mu \right)\left(1-2\mu \right)}}\\ C_{T}=\sqrt{\frac{E}{2\rho \left(1+\mu \right)}} \end{array}$$
where *ρ is* the density; *E* is the Young’s modulus; *μ* is the Poisson ratio; and *C_L_* and *C_T_* are the velocity of the longitudinal and the transverse waves, respectively.

With the knowledge of the distance between the piezoelectric aggregates and the wave propagation velocity, the time of arrival can be estimated. By comparing the estimated time of arrival with the wave peaks in the acquired signals, the wave packets of the longitudinal and transverse waves can be recognized and differentiated. In the cement medium, the longitudinal waves propagate faster than the transverse waves. Therefore, the first wave peak in the acquired signal corresponds to the longitudinal wave, while the second wave peak corresponds to the transverse wave. 

Due to the wave scattering at the interface of the damage, the elastic waves attenuate. By analyzing the energy loss of the transmitted waves, it is possible to quantify the severity of the damage. However, since the cement materials contain various hidden gravels and pores, countless wave diffraction and reflection occur at the interfaces of the discontinuities, causing complicated waveforms in the acquired signals. Since the dimensions of the discontinuities are much smaller than that of the macro damage, the wave scattering at the discontinuity is more likely to affect the high-frequency components in the acquired signal. Therefore, it is necessary to eliminate the high-frequency components, and thus to extract the low-frequency components, which have a strong correlation with the macro damage. A feasible way to fulfill this goal is to perform a multi-scale signal decomposition. Therefore, the wavelet packet analysis is adopted to decompose the transmitted waves to a set of wave components with different frequencies. By analyzing the energy loss of each wave component, a damage quantification index can be intentionally formulated.

### 2.2. The Wavelet Packet-Based Energy Analysis

Since the wavelet packet analysis can effectively differentiate the wave components with different frequencies and facilitate the formulation of the damage quantification index, the basic procedure of the wavelet packet-based energy analysis is briefly reviewed in this section.

The wavelet packet analysis can decompose the elastic wave signals into the approximations (low-frequency components) and the details (high-frequency components), as shown in Figure 1. For a wave signal ***X***, the n-level wavelet packet decomposition can produce 2*^n^* signal components as [*X*_1_, *X*_2_, …, *X*_2_*^n^*].

For the signal component at a specific level *j*, the *X_j_* can be expressed as
(2)$$X_{j}=[X_{j,1},X_{j,2},\ldots ,X_{j,m}]$$
where *j* is the frequency band (j = 1, 2, …, 2*^n^*) and *m* is the sampling number.

The energy of each signal component can be calculated as
(3)$$E_{j}=\overset{k=m}{\underset{k=1}{\sum }}X_{j,k}$$

Accordingly, the energy of the wave signal ***X*** can be calculated by summing the energy of each signal component as
(4)$$E=\overset{l=j}{\underset{l=1}{\sum }}X_{l}$$

By intentionally select the first *j* signal components, the signal components with high frequencies can be eliminated, presenting the total energy of the low-frequency components. With this implementation, a damage quantification index can be formulated to characterize the energy variation of the low-frequency components, which is highly affected by the severity of the macro damage. In this study, the wave signals acquired from both the numerical and the experimental investigations are decomposed by the abovementioned procedures, generating a set of wave components. The mother wavelet is set to the Harr wavelet, and the decomposition level is set to three. The wavelet packet decomposition is coded into a Matlab program, and the energy of the wave components of the first three levels are added up to form a damage quantification index as shown in Equation (4).

## 3. Improved Piezoelectric Aggregate

The traditional piezoelectric aggregate (TPA) are normally sealed with cement blocks. However, the water molecule inside the cement paste cannot be completely excluded during this process. Due to the electrical conductivity of the water molecule, the TPA cannot insulate the electric current from the testing equipment. In some cases, the cement specimen may act as a conductor and form a short circuit. The electric pulse sent from the wave generator may pass through the specimen and directly arrive at the wave receiver, causing serious cross talk. Since this cross talk comes in the form of electric currents, its amplitude is much higher than the electric current generated by the piezoelectric effect. Some published studies have reported serious cross talk in the signals acquired by the TPA [27]. 

However, since the overvoltage protection inserts some additional resistance inside the testing equipment to prevent the high voltage from burning the equipment, the presence of the cross talk may cause the loss of the signal details with low voltage. Worse, the signal details with low voltage are the actual response caused by the elastic waves, which contain valuable information regarding to the damage severity. The loss of the signal details may impair the accuracy and robustness of the damage quantification. To alleviate the cross talk, an improved piezoelectric aggregate (IPA) is developed in this section. The signals acquired from a confirmatory experiment indicate that the signals acquired by the IPA have less cross talk than that by the TPA.

### 3.1. Preparation of the TPA

The piezoelectric ceramics can generate electric fields under the external stress or strain, which is called the direct piezoelectric effect. On the other hand, the piezoelectric ceramics can generate stress or strain under an external electric field, which is called the converse piezoelectric effect. Due to the piezoelectric effect, the piezoelectric ceramics can act as both the actuator and the sensor. Especially, the Lead Zirconate Titanate (PZT) gets certain popularity because of its linear piezoelectric effect. However, the PZTs are very fragile and may break during the cement casting process.

In order to protect the PZT, the traditional way is to seal the PZT patch with the cement paste to form a piezoelectric aggregate. At first, two wires are welded on the surfaces of the PZT patch (See Figure 2a). Then, the PZT patch is placed in the hole of a steel mold, which is half-filled with the cement paste, and the wire is embedded in the groove (See Figure 2b). Next, more cement paste is impounded to fill the hole completely. Finally, the aggregate can be removed by a rubber hammer when the cement paste gets initially set. The basic preparation procedure of the TPA is shown in Figure 2.

### 3.2. Preparation of the IPA

To alleviate the cross talk, a new aggregate packing method is developed to form the IPA. First, a Polymethyl Methacrylate (PMMA) tube is made and a hole is drilled on the tube wall to allow the wire to pass through. Then the piezoelectric ceramic is placed in the midsection of the tube, and epoxy resins are impounded into the tube. Finally, the aggregate is pushed out after the epoxy resin comes to a complete set. Since the epoxy resin is able to completely insulate the electric currents in the cement specimen, the cross talk in the acquired signal can be effectively alleviated. The basic preparation procedure of the IPA is shown in Figure 3.

### 3.3. Comparison between the IPA and the TPA

To test the performance of the IPA and the TPA, a set of measurements is conducted on Portland cement beams with the dimension of 500 × 100 × 100 mm, as shown in Figure 4. The curing parameters and the material properties after the final set of the cement beams are given in Table 1.

Meanwhile, the IPAs and the TPAs are prepared according to the manufacture methods described above, and the dimensions of the aggregate are given in Table 2. The material properties of the epoxy resin are given in Table 3.

Two aggregates are embedded at the ends of the beam, functioning in the pitch-catch mode. The elastic wave is excited by the left aggregate and then acquired by the right aggregate. The incident wave is set to a sinusoid function with a central frequency of 90 kHz. The distance between the actuator and the receiver is set to 460 mm.

The signals acquired by the IPA and the TPA are shown in Figure 5. It can be seen that the signal acquired by the TPA suffers serious cross talk near 0.04 ms, and the wave peak of the cross talk owns a much higher amplitude than that caused by the actual elastic waves. Due to the high electric resistance of the epoxy resin, the signals acquired by the IPA suffer less cross talk than that by the TPA and thus record accurate signal details.

Additionally, the frequency responses of the excitation and the signals acquired by the IPA and the TPA are calculated through the Fast Fourier Transform (FFT). The energy distribution with respect to the frequency is shown in Figure 6. The FFT spectrum of the TPA reports a fluctuant energy peak. This fluctuation may impair an equal acquisition for all the signal components with different frequencies. Some signal components with the frequency of low energy may be drowned out by the others. In contrast, the frequency response of the IPA is close to that of the excitation. A smooth wave peak is observed in the frequency response of the IPA, which means the IPA can acquire different signal components with similar energy.

## 4. Numerical Investigation

### 4.1. Expilict Finite Element Analysis

For a system with multi-degrees of freedom, the basic dynamic equilibrium equation is given as
(5)$$M\ddot{U}+C\dot{U}+KU=P$$
where ***M*** is the mass matrix, ***C*** is the damping matrix, ***K*** is the stiffness matrix, ***P*** is the vector of the external loads, $\ddot{U}$ is the acceleration, $\dot{U}$ is the velocity, and ***U*** is the displacement. The relationship of displacement, velocity, and acceleration can be expressed as
(6)$$\begin{array}{l} {\ddot{U}}^{\left(t\right)}=\frac{1}{{\Delta t}^{2}}\left(U^{\left(t-\Delta t\right)}-2U^{\left(t\right)}+U^{\left(t+\Delta t\right)}\right)\\ {\dot{U}}^{\left(t\right)}=\frac{1}{{2\Delta t}^{2}}\left(-U^{\left(t-\Delta t\right)}+U^{\left(t+\Delta t\right)}\right) \end{array}$$
where Δ*t* is the time increment. Therefore, the displacement $U^{\left(t+\Delta t\right)}$ in the time period *t +* Δ*t* can be solved through
(7)$$\left(\frac{1}{{\Delta t}^{2}}M+\frac{1}{2\Delta t}C\right)U^{\left(t+\Delta t\right)}=R^{\left(t\right)}-\left(K-\frac{2}{{\Delta t}^{2}}M\right)U^{\left(t\right)}-\left(\frac{1}{{\Delta t}^{2}}M-\frac{1}{2\Delta t}C\right)U^{\left(t-\Delta t\right)}$$

Since the displacement $U^{\left(t+\Delta t\right)}$ is based on the equilibrium, the integration procedure is called the explicit method. To increase the computational efficiency of the explicit integration method, a diagonal mass matrix is used for each finite element:(8)$${\ddot{U}}^{\left(i\right)}={\left[M\right]}^{-1}\left({\left[F\right]}^{\left(i\right)}-{\left[I\right]}^{\left(i\right)}\right)$$
where [***F***] is the external force vector, [***I***] is the internal force vector, and [***M***]^−1^ is the inversed mass matrix.

When modeling the elastic waves, the time increment and the element size need to be carefully selected. To ensure a stable numerical iteration, the time increment should be less than the critical value of
(9)$$\Delta t=\min\left(\frac{L_{e}}{c}\right)$$
where *c* is the elastic wave velocity and *L_e_* is the smallest characteristic dimension of the finite element. Meanwhile, to get a sufficient accuracy, at least seven finite elements should be set within each wavelength.

### 4.2. Numerical Modelling

A finite element model is built in the Abaqus explicit based on the explicit method described above. The wave excitation, propagation, and scattering processes are numerically simulated in a cement beam. The dimensions of the cement beam are set to 500 × 100 × 100 mm, and the density, elastic modulus, and Poisson’s ratio are set to 2350 kg/m^3^, 3.25 × 10^10^ Pa, and 0.2, respectively. Two piezoelectric aggregates are embedded along the longitudinal axis (20 mm from the end). The piezoelectric aggregates are idealized as the mesh nodes to excite or receive the elastic waves. A perpendicular crack is applied at the mid-span of the beam to simulate the damage due to bending. The width and the length of the crack are set to 2 mm and 100 mm, respectively. The crack depth ranges from 0 to 55 mm with increment of 5 mm each time. The configuration of the finite element model is shown in Figure 7. Considering the numerical stability and the computational efficiency, the time increment Δ*t*, the element length Δ*x*, and the element type are set to 1 × 10^−8^ s, 1 mm, and C3D8R, respectively. 

The excitation is set to a tone burst, which is a 6-cycle Hanning windowed sinusoid signal:(10)$$x\left(t\right)=\sin\left(2\pi f_{c}t\right)\left[1-\cos\left(\frac{2\pi f_{c}t}{6}\right)\right]$$
where *f_c_* is the central frequency, which is 90 kHz in this study. The left aggregate acts as the actuator, and the right aggregate acts as the receiver. The excitation is applied as nodal displacements at the left aggregate. The elastic waves propagate from left to right, passing through the crack at the mid-span. Due to the wave scattering at the interface of the crack, the energy of the transmitted waves decreases. The snapshots of the wave propagation in the beams with different crack depths are shown in Figure 8, Figure 9 and Figure 10, respectively. The Mises stresses in these snapshots depict the excitation, propagation, scattering, and reception of the elastic waves at different time steps. Due to the growth of the crack depth, less elastic waves can pass through the mid-span and arrive at the right end. Therefore, by analyzing the wave energy received by the right aggregate, the crack depth can be quantified based on the wavelet packet energy analysis.

The received signals are extracted as the nodal displacement at the right aggregate. To ensure an accurate acquisition of the elastic waves, the sampling frequency is set to 100 MHz. The received signals are successively imported to a precoded Matlab program to calculate the WPEIs.

### 4.3. Damage Quantification Based on the Simulated Data

The transmitted waves acquired by the right piezoelectric aggregate are shown in Figure 11. In accordance with the snapshots in Figure 8, Figure 9 and Figure 10, the amplitude of the transmitted wave decreases with the growth of the crack depth.

The signals collected from the different cases are analyzed using the wavelet packet-based energy analysis method presented above. The simulated signals are decomposed through the wavelet packet decomposition. The mother wavelet is set to the Harr wavelet, and the decomposition level is set to three, producing eight signal components. No wave filtering or intentional selection of particular signal components are performed during this process. The energy of the signal components of the first three levels are added up according to Equation (4), generating the wavelet packet-based energy indices (WPEI) as shown in Figure 12. The results indicate that the WPEI is inversely proportional to the crack depth. The regression analysis is performed on the WPEIs to characterize the linearity of the WPEI. The variation of the WPEI can be expressed by a linear function as
(11)$$\mathrm{WPEI}=-1.01\times 10^{-10}\times d+1.01\times 10^{-8}$$
where *d* is the crack depth.

A set of parameters, such as the residual sum of squares, adjusted R-square, and standard errors of the intercept and slope, are presented to evaluate the fitness of the linear approximation. The results, as shown in Table 4, demonstrate that the linear expression can accurately depict the variation of the WPEI with respect to the crack depth. The linear relationship between the WPEI and the crack depth can serve as a reference to estimate the crack depth according to the WPEI. The results demonstrate that the WPEI is a reliable damage quantification parameter.

## 5. Experimental Investigation

### 5.1. Experimental Setup

In this section, the crack depth at the mid-span of a cement beam is quantified based on the wavelet packet-based energy analysis. 

The IPAs are prepared as described in Section 3.2. The picture of the IPA can be referred to in Figure 3c, and the dimension and the material properties are given in Table 2 and Table 3, respectively. Two IPAs are embedded at the ends of the beam, acting as the actuator and the receiver. The piezoelectric aggregates are fixed by iron wires at predetermined locations. The configuration of the piezoelectric aggregates and the dimension of the beam are the same as the numerical model. By embedding the IPAs, the elastic waves can be excited and received in real time, forming a structural health monitoring system for the cement beam. Abnormalities in the received signals can indicate the occurrence or the growth of the damage. Furthermore, with these embedded IPAs, the physical parameters of the locations deep inside the structure can be acquired, fulfilling a global damage identification.

The setup of the experiment is shown in Figure 13. The D-Space data acquisition system is employed to coordinate the wave excitation and reception. To acquire at least 10 samples per wavelength, the sampling rate of the D-Space system is set to 200 kHz. A tone bust, which is a harmonic sinusoid with a frequency of 90 kHz, is generated by a laptop and sent to the D-Space system. The D-Space system can transform the digital signal to the electric voltage and excite the piezoelectric aggregate to generate elastic waves. Since the propagation of the elastic waves in the cement medium suffers serious attenuation, a charge amplifier is used to amplify the signals. The transmission gain of the charge amplifier is set to 100 mv/pc.

The cement beams with different crack depths are tested. Artificial cracks are perpendicularly cut at the mid-span, with the depth varying from 0 to 55 mm with increment of 5 mm each time (see Figure 14). For each crack depth, five separate measurements are conducted to produce an average signal. With this implementation, the interference of uncertain factors in the measurement can be alleviated. The baseline is acquired as the signal from an intact cement beam to set a reference for the subsequent measurements. 

### 5.2. Damage Quantification Based on the Measured Data

The signals acquired from the beam with the growing crack depth are shown in Figure 15. Although the energy of the acquired signals roughly decreases with the growth of the crack depth, it is difficult to estimate the crack depth directly according to the acquired signals in the time domain. Since there is no wave filtering or de-noising process adopted in this experiment, the signals in Figure 15 contain a certain amount of noise. The presence of the noise affects the quality of the signal. Some differences can be observed by comparing the simulated signals (see Figure 11) to the measured signals (see Figure 15).

The signal energy is transformed to the WPEI based on the wavelet packet analysis mentioned above. The measured signals are decomposed through the wavelet packet decomposition. The mother wavelet is set to the Harr wavelet, and the decomposition level is set to three. Eight signal components are produced by this process. The energy of the eight signal components are added up according to Equation (4). The WPEIs from different scenarios are shown in Figure 16. It is clearly seen that the WPEI from the intact beam reports the highest value, and the WPEI gradually decreases as the crack depth increases.

By applying the regression analysis, it can be seen that the WPEI exhibits strong linearity compared to the signals in the time domain, as shown in Table 5. The results of the linear regression show that the variation of the WPEI is linear. According to the value of the WPEI, the crack length can be estimated through a linear function as
(12)WPEI=−0.0033×d+0.4529

## 6. Conclusions and Path Forward

Based on theoretical, numerical, and experimental analyses, the crack at the mid-span of the cement beam can be quantified based on the energy of the transmitted elastic waves. The elastic waves can be robustly excited and acquired by the piezoelectric aggregates and thus can fulfill the active structural health monitoring for cement-based structures. Based on the results revealed by this study, some conclusions and the path forward can be given as follows.

### 6.1. Conclusions

The IPA sealed by the epoxy resin has better electric insulation than that of the TPA sealed by the cement. The new sealing process can effectively alleviate the cross talk in the acquired signal and facilitate the signal interpretation and damage quantification.The wavelet packet analysis is a reliable method to extract the elastic wave energy. The damage quantification index proposed in this paper is able to provide a linear characterization of the crack depth at the mid-span of the cement beam. According to the linear expression from the linear regression, the crack depth can be estimated by referring to the damage quantification index (WPEI).The embedment of the IPA can fulfill a global structural health monitoring for cement-based structures. Due to the good strength, high electric resistance, and compatible deformation, the IPAs can be built inside the structure and can characterize the structural state in real time.

### 6.2. Path Forward

The piezoelectric aggregates require wires to provide power to excite the elastic waves, causing wiring issues. Self-powered or wireless powered piezoelectric aggregates can highly promote the application of the piezoelectric aggregates.The deformation of the piezoelectric aggregates can excite elastic waves in all directions. The diverging propagation of the elastic wave may cause energy loss and multiple reflections at the structural boundary, obstructing the interpretation of the acquired signals. A directional transducer, which can focus waves in a particular direction, is desirable to enhance the detection range.

## Figures and Tables

**Figure 1 sensors-19-00425-f001:**
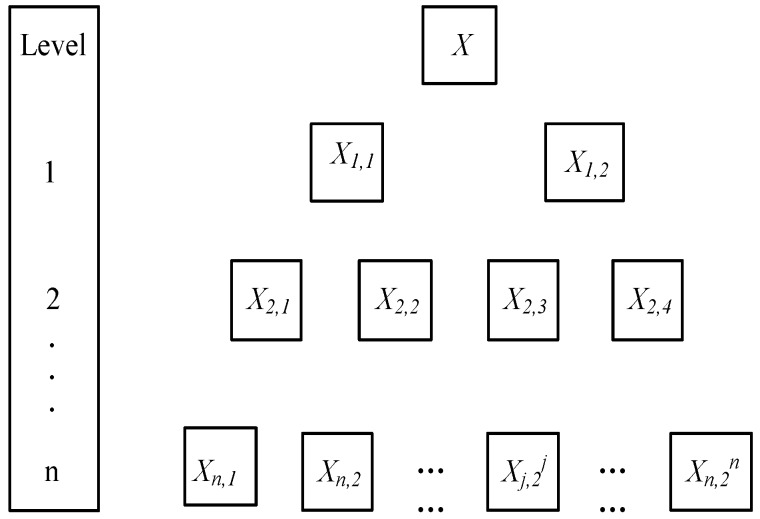
Decomposition tree of the wavelet packet analysis.

**Figure 2 sensors-19-00425-f002:**
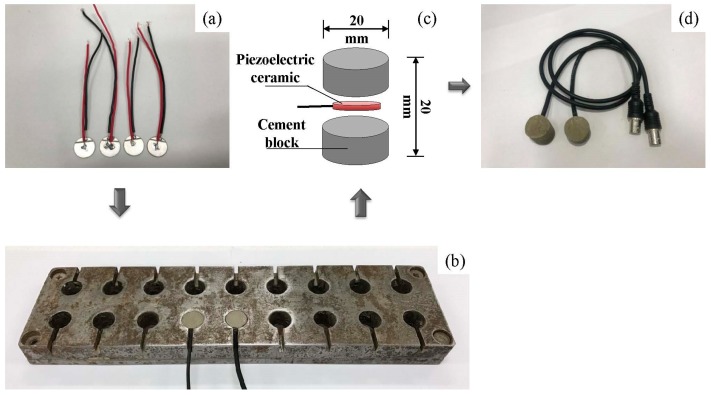
The manufacturing process of the traditional piezoelectric aggregate (TPA): (**a**) the Lead Zirconate Titanate (PZT) patches; (**b**) the steel mold; (**c**) the configuration of the TPA; and (**d**) the finished TPA.

**Figure 3 sensors-19-00425-f003:**
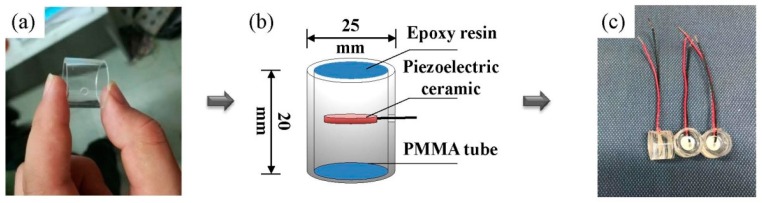
The manufacturing process of the improved piezoelectric aggregate (IPA): (**a**) the Polymethyl Methacrylate (PMMA) tube; (**b**) the configuration of the IPA; and (**c**) the finished IPA.

**Figure 4 sensors-19-00425-f004:**
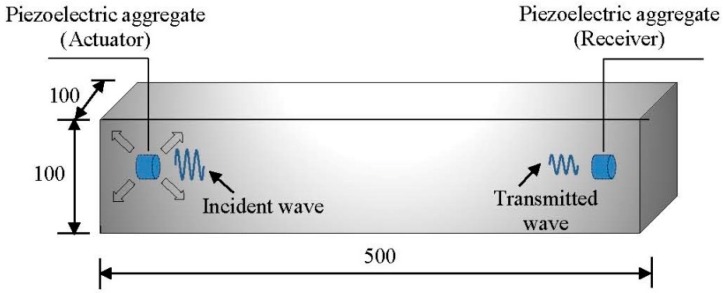
The pitch-catch mode to compare the signals of different piezoelectric aggregates.

**Figure 5 sensors-19-00425-f005:**
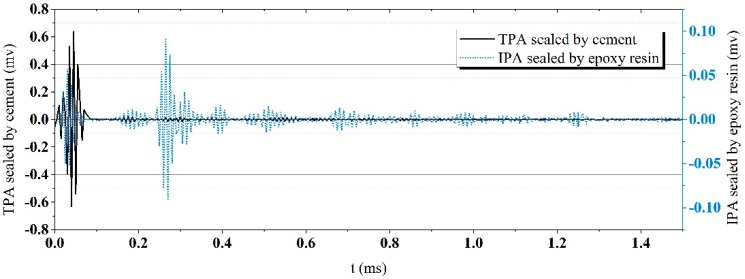
Signals acquired by different piezoelectric aggregates.

**Figure 6 sensors-19-00425-f006:**
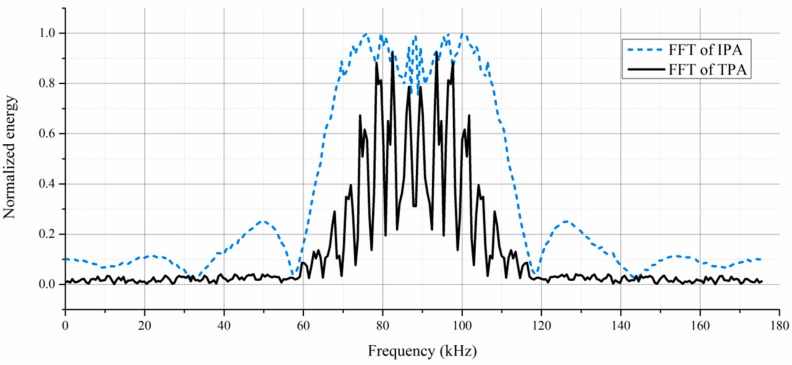
Frequency responses of the IPA and the TPA.

**Figure 7 sensors-19-00425-f007:**
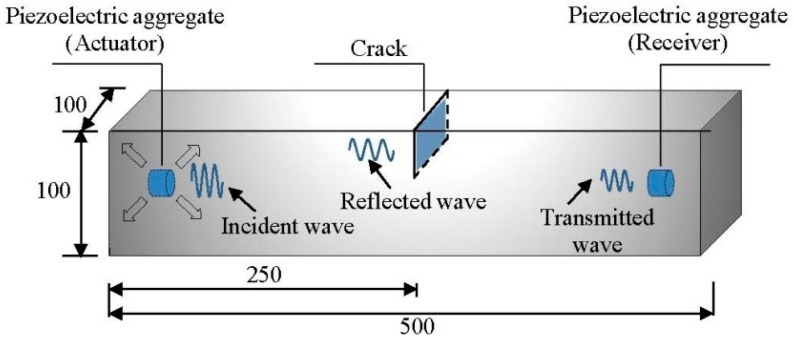
Configuration of the finite element model.

**Figure 8 sensors-19-00425-f008:**
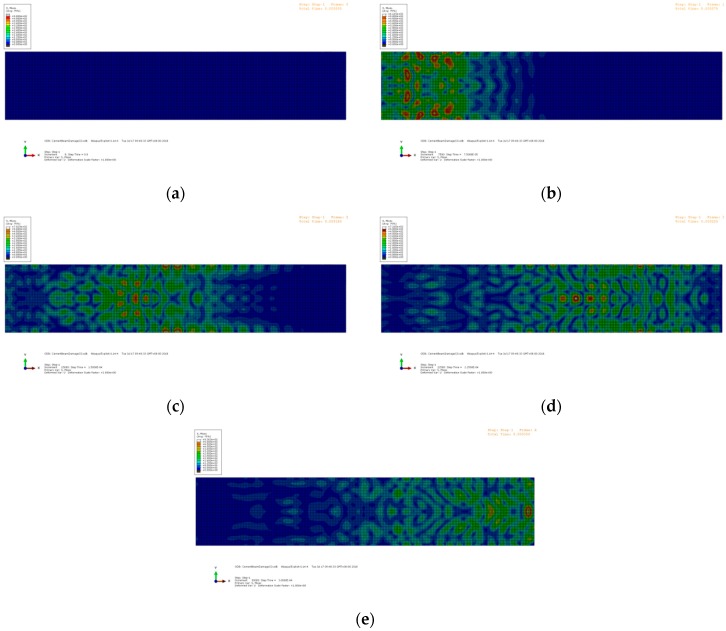
Snapshots of the numerical model with the crack depth = 0 mm: (**a**) t = 0 ms; (**b**) t = 75 ms; (**c**) t = 150 ms; (**d**) t = 225 ms; and (**e**) t = 300 ms.

**Figure 9 sensors-19-00425-f009:**
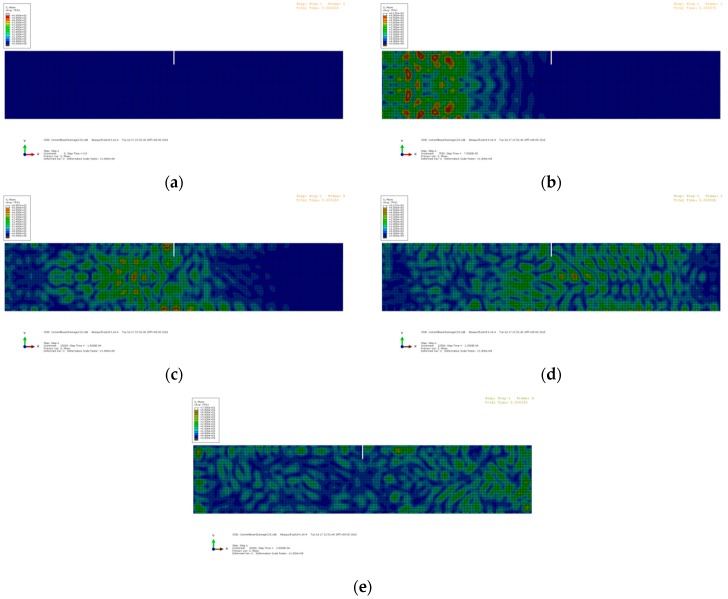
Snapshots of the numerical model with the crack depth = 20 mm: (**a**) t = 0 ms; (**b**) t = 75 ms; (**c**) t = 150 ms; (**d**) t = 225 ms; and (**e**) t = 300 ms.

**Figure 10 sensors-19-00425-f010:**
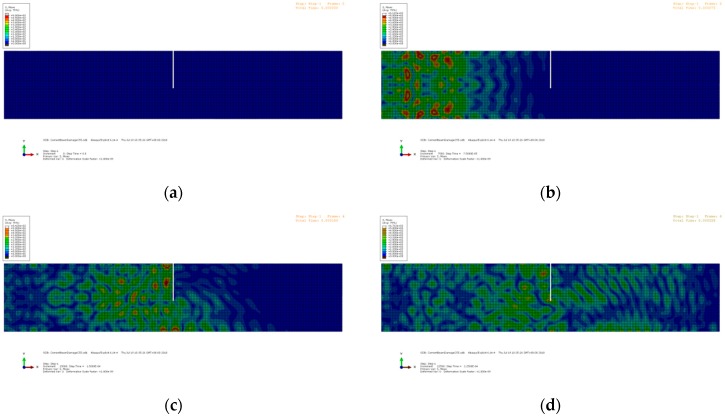
Snapshots of the numerical model with the crack depth = 55 mm: (**a**) t = 0 ms; (**b**) t = 75 ms; (**c**) t = 150 ms; (**d**) t = 225 ms; and (**e**) t = 300 ms.

**Figure 11 sensors-19-00425-f011:**
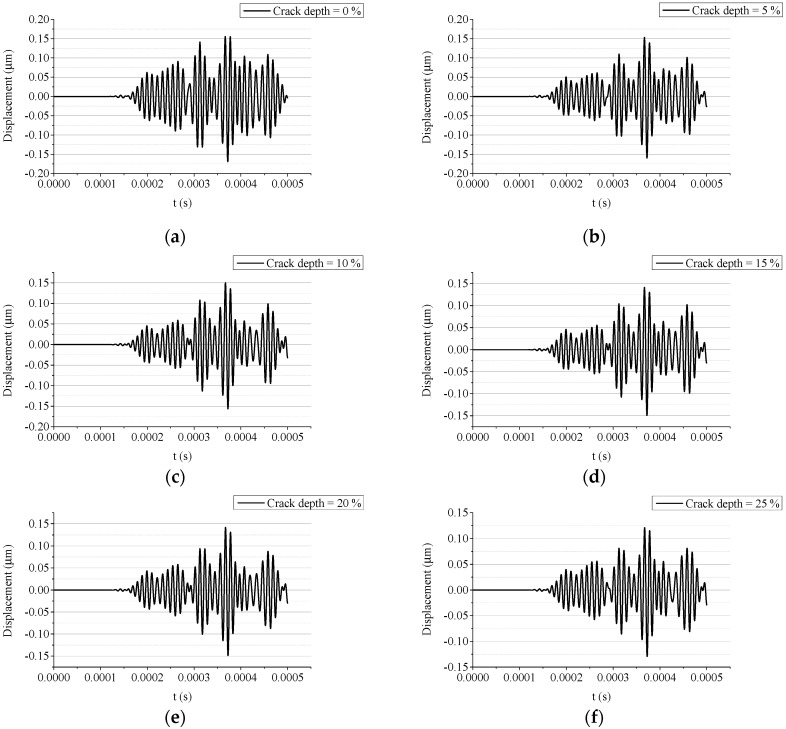
Acquired signals from simulations with different crack depths: (**a**) crack depth = 0 mm (0%); (**b**) crack depth = 5 mm (5%); (**c**) crack depth = 10 mm (10%); (**d**) crack depth = 15 mm (15%); (**e**) crack depth = 20 mm (20%); (**f**) crack depth = 25 mm (25%); (**g**) crack depth = 30 mm (30%); (**h**) crack depth = 35 mm (35%); (**i**) crack depth = 40 mm (40%); (**j**) crack depth = 45 mm (45%); (**k**) crack depth = 50 mm (50%); and (**l**) crack depth = 55 mm (55%).

**Figure 12 sensors-19-00425-f012:**
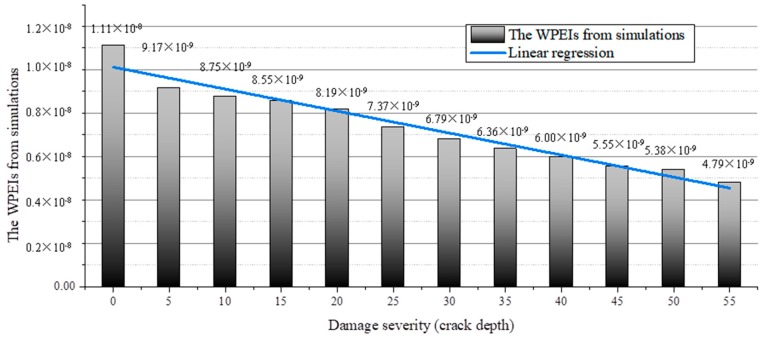
Linearity of the WPEI with respect to the crack depth (numerical investigation).

**Figure 13 sensors-19-00425-f013:**
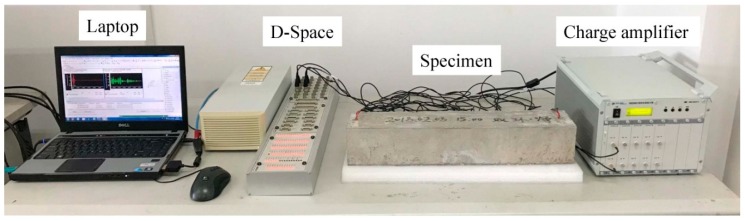
Setup of the testing equipment.

**Figure 14 sensors-19-00425-f014:**
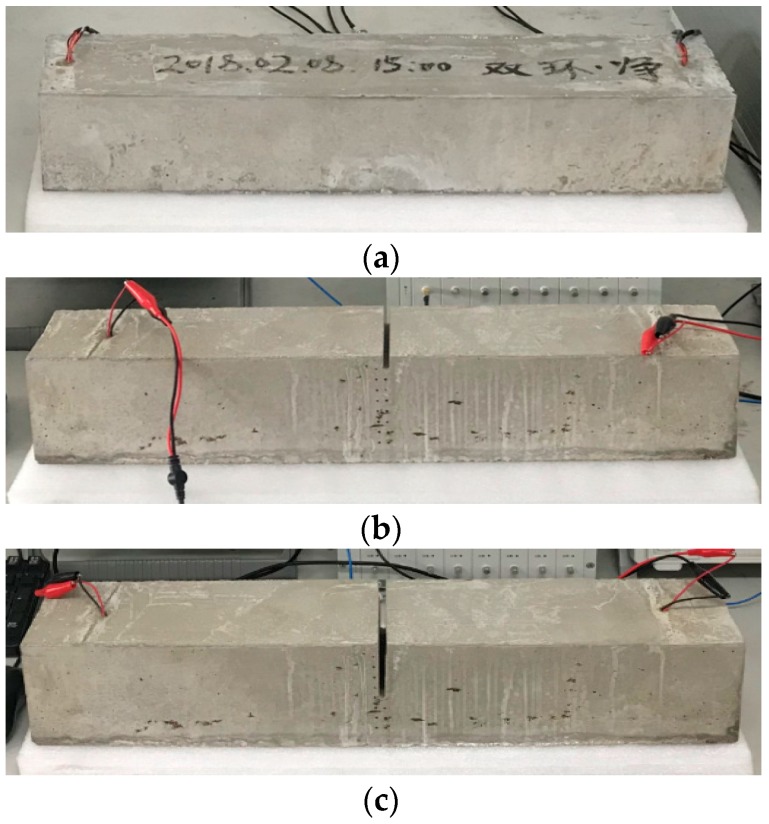
The specimens of the cement beams with different crack depths: (**a**) crack depth = 0 mm; (**b**) crack depth = 20 mm; and (**c**) crack depth = 55 mm.

**Figure 15 sensors-19-00425-f015:**
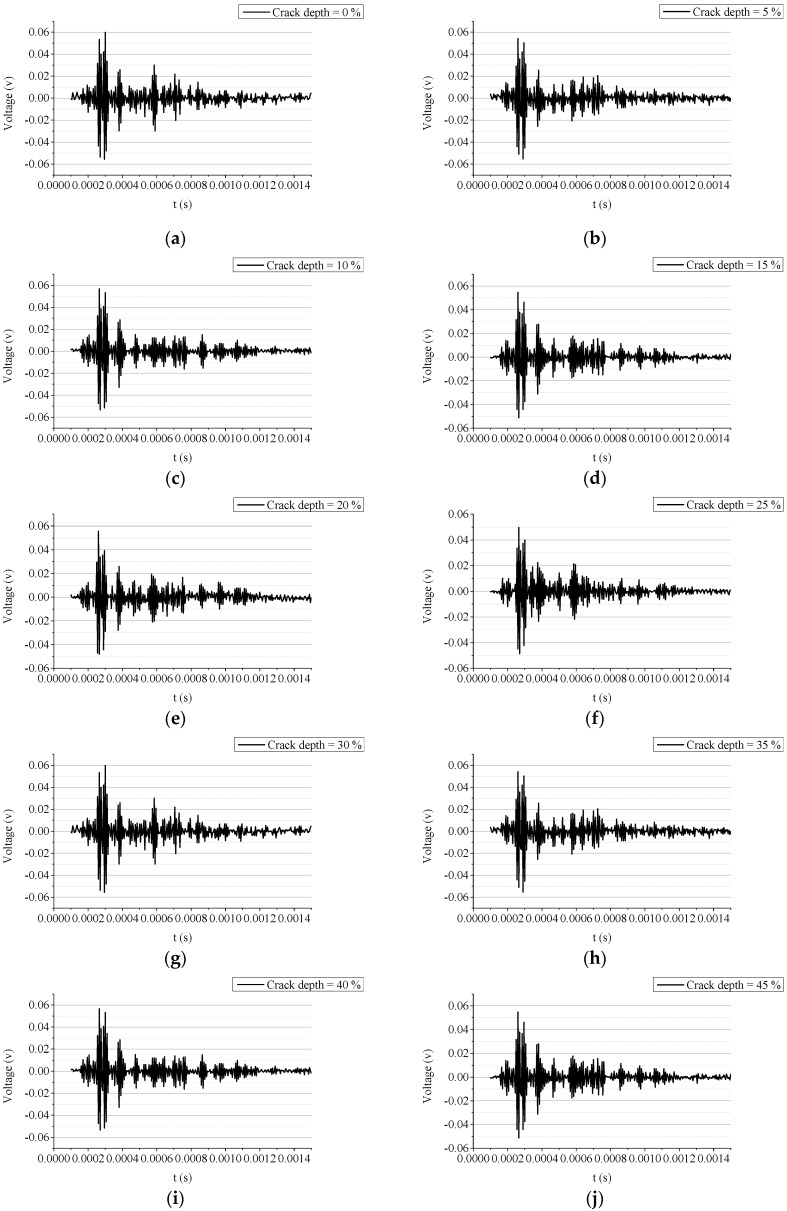
The acquired signals from the experiments with different crack depths: (**a**) crack depth = 0 mm (0%); (**b**) crack depth = 5 mm (5%); (**c**) crack depth = 10 mm (10%); (**d**) crack depth = 15 mm (15%); (**e**) crack depth = 20 mm (20%); (**f**) crack depth = 25 mm (25%); (**g**) crack depth = 30 mm (30%); (**h**) crack depth = 35 mm (35%); (**i**) crack depth = 40 mm (40%); (**j**) crack depth = 45 mm (45%); (**k**) crack depth = 50 mm (50%); and (**l**) crack depth = 55 mm (55%).

**Figure 16 sensors-19-00425-f016:**
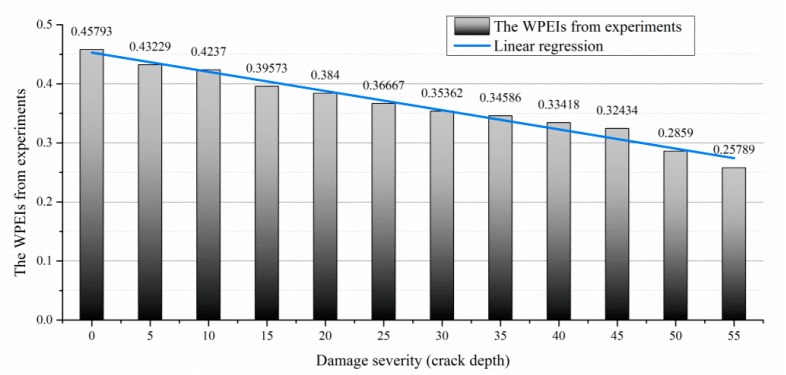
Linearity of the WPEI with respect to the crack depth (experimental investigation).

**Table 1 sensors-19-00425-t001:** Curing parameters and material properties.

**w/c Ratio**	**Curing Period**	**Curing Temperature**	**Curing Moisture**
0.36	28 days	20 ± 2 °C	97 ± 2%
**Density**	**Poisson Ratio**	**Elastic Modulus**	**Electric Resistance**
2350 kg/m^3^	0.2	3.25 × 10^10^ Pa	6.00 × 10^9^ Ω·m

**Table 2 sensors-19-00425-t002:** Dimensions of the TPA and the IPA.

Size (mm)	TPA	IPA	Piezoelectric Ceramic
Inner diameter	20	20	14
Height	20	20	1

**Table 3 sensors-19-00425-t003:** Material properties of the epoxy resin.

Density	Poisson Ratio	Elastic Modulus	Electric Resistance
1200 kg/m^3^	0.38	2.00 × 10^9^ Pa	1.60 × 10^14^ Ω·m

**Table 4 sensors-19-00425-t004:** Results of the linear regression (numerical investigation).

Residual Sum of Squares	Adjusted R-Square	Intercept (Standard Error)	Slope (Standard Error)
1.6693 × 10^−18^	0.9522	1.0124 × 10^−8^ (2.2186 × 10^−10^)	−1.0142 × 10^−10^ (6.8333 × 10^−12^)

**Table 5 sensors-19-00425-t005:** The results of the linear regression (experimental investigation).

Residual Sum of Squares	Adjusted R-Square	Intercept (Standard Error)	Slope (Standard Error)
9.3882 × 10^−4^	0.9733	0.4529 (0.0053)	−0.0033 (1.6205 × 10^−4^)

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
