# Peer review of "Damage Quantification with Embedded Piezoelectric Aggregates Based on Wavelet Packet Energy Analysis"

_sensors, 2019, doi:10.3390/s19020425_

Round 1

Reviewer 1 Report

The submitted research article “sensors-431579” entitled: “Damage quantification with embedded piezoelectric aggregates based on wavelet packet-based energy analysis” contains original contribution to the implementation of embedded piezoceramic materials for the damage diagnosis and cracking of concrete elements. The application of low cost, active sensing, embedded Piezoelectric lead Zirconate Titanate (PZT) transducers as Smart Aggregates (SA) has been proved a promising Structural Health Monitoring (SHM) technique that recently has been applied in several Reinforced Concrete (RC) members. The use of the examined technique as an alternative diagnostic system based on the energy of the transmitted elastic waves reveals an interesting perspective that is still open to question. In the submitted paper a wavelet packet analysis is adopted to calculate the energy of the transmitted elastic waves based on Improved Piezoelectric Aggregates (IPAs) that have been developed in order to detect concrete cracking. Damage quantification index values of the IPAs signals were used to identify the different crack depths at the mid-span of a set of cement beam specimens. The study includes an experimental and an analytical investigation. The specific tasks of the presented study are clear. The manuscript is very interesting, well-structured and well-presented. It also presents an interesting study that derives new results in a clear and organized manner. As an overall comment, the article could be published after revision based on the following recommendations:

1.  It would be very useful to provide the original frequency response signals of more IPAs than those illustrated in Fig. 5.

2.  Details of the used IPAs should be highlighted along with the superiority of the proposed SHM technique in the experiments. Further, come comments concerning the feasibility of this method on the state of the practice could be reported.

3.  The frequency range in which the response signals used to calculate damage index should further be addressed and justified. The used damage index also seems to depend on a single frequency band, and occasionally cannot accurately reflect the development of structural damage. Some comments on this issue should be added.

4.  Although the literature review is rather informative, the following relevant issues have not been considered, although that they will promote the research significance of this study:

(a)   The use of post-installed surface-mounted (externally epoxy bonded) PZTs or pre-installed embedded PZTs (acting as smart aggregates) has recently been extended, not only for the damage evaluation of cement beams (or plain concrete elements), but also for damage detection/identification, the assessment of their severity level and even more the on-line monitoring of the possible damage evolution with time in RC structural members under shear/flexural monotonic and cyclic/seismic loading using the well-known Electro-Mechanical Admittance (EMA) technique.

(b)   The recent developments of portable, real-time, wireless impedance/admittance SHM systems which have easily and successfully applied to large-scaled RC members of existing RC structures under monotonic and cyclic laoding.

(c)   The implementation of a network of small-sized PZTs, instead of individual (single) transducers, provides more accurately results concerning the evaluation of concrete strength gain characteristics and the diagnosis of damages in RC structural members. It is emphasised that the presented work also implies the installation of a series of PPTs acting as sensors and an additional network of PPTs acting as actuators. The signal measurements of a series of surface-mounted or internally embedded PZTs arrayed in specific locations of the examined concrete member improve the damage assessment procedure providing better correlation of the local damage level and the values of the used damage index. Especially in cases of shear-critical RC members, which exhibit brittle sudden diagonal cracking, the damage index values calculated by the network of the installed PZTs lead to a safe determination of the locus and the magnitude of the occurred damage at different levels prior the fatal failure (early damage stages).

It is obvious that these issues are quite relative to the submitted paper and, therefore, it is strongly suggested the findings of the following additional references (ordered by date) to be considered in order to further establish the research significance and to promote the objectives of this study:

-    Numerical and experimental studies on damage detection of concrete beam based on PZT admittances and correlation coefficient, Construction and Building Materials, 2013.

-    Combined embedded and surface-bonded piezoelectric transducers for monitoring of concrete structures, NDT&E Int 2014.

-    Damage evaluation in shear-critical reinforced concrete beam using piezoelectric transducers as smart aggregates, Open Engineering, 2015.

-    Detection of flexural damage stages for RC beams using piezoelectric sensors (PZT), Smart Structures and Systems, 2015.

-    Experimental Application of a Wireless Earthquake Damage Monitoring System (WiAMS) using PZT Transducers in Reinforced Concrete Beams, WIT Trans. The Built Env., 2015.

-    Investigation of a new experimental method for damage assessment of RC beams failing in shear using piezoelectric transducers, Engineering Structures, 2016.

-    Applications of smart piezoelectric materials in a wireless admittance monitoring system (WiAMS) to structures - Tests in RC elements, Case Studies in Construction Materials, 2016.

-    Mechanical impedance based embedded piezoelectric transducer for reinforced concrete structural impact damage detection: A comparative study, Construction and Building Materials, 2018.

Author Response

Please see the reply in the attached file.

Reviewer 2 Report

This study measured the degree of damage associated with improved smart piezoelectric aggregate by reducing cross talk through PMMA and epoxy resins and shows the damage quantifications by converting energy received by ultrasonic wave into WPEI. Although, there seems no logical problem, it is hard to find new contribution compared to existing research. It is necessary to reflect the followings.

1. line 34-55: minimum reference should be added.

2. line 76-116: It is difficult to find difference from damage evaluation through typical WPEI associated with the smart piezoelectric aggregate in previous other researches. The authors should explain the difference with other studies.

3. line 132:  What is difference between damage severity and damage quantification? Explanation should be necessary.

4. In sections 2.2, 4.2 and 5.2, the principle of WPEI, the used program, and the process details should be explained.

5. In figure 5, the improvement of smart piezoelectric aggregate sealed by epoxy resin seems to be unclear. Why the first black line group is larger than the blue line? The authors should explain this graph logically.

6. Overall, it is not a new fact or a new theory & method that addresses the issue of quantifying damage evaluation with measured energy by improving the existing smart piezoelectric aggregate.

Absolutely, as the crack depth increases, the received energy decreases. This paper should propose a new method for measuring outer or inner cracks, voids, and material properties reduction in the field concrete structures. Additionally, new approach to evaluate the actual damage with various variables such as filtering, arrival time and energy, and frequency range change. The authors should suggest a new approach to identify the characteristics of ultrasound sensitive to the damage of the concrete inside and outside.

Author Response

(The authors gave the same response as above.)

Reviewer 3 Report

In the last decade,  there were several groups  who development PZT based embedding technique.  In order to attract readers from other groups it is better to cite articles of electromechanical Impedance,  ultrasonic etc who use similar  PZT pulse echo or PZT impedance. Thus, you will attract reaserchers from multiple areas (pzt based).

  Especially cite at least 5 to 8 articles of pzt emeddedment in last decade. Your paper is good but due credit is needed even for papers published a few years ago. 

   What is the maximum resistance  your embedded sensor can offer? Will your result change if PZT size changes?

 Is there any noise observed in raw signals? Can it work in harsh environments?

Author Response

(The authors gave the same response as above.)

Reviewer 4 Report

I find this paper interesting and worth publishing. Maybe it would be worth to incorporate into the text some analysis of the influence of the bond/glue on the signal transmission from the PZT into material? and there further into the analysed specimen? Is there any damping?

Fig. 10 and 14 - It would be beneficial to the paper to have as a result of analysis performed a relative size of crack instead of actual depth in mm. Besides including 12 different signals in one picture is hard to recognise anyway. Maybe somehow divide the pictures?

Author Response

(The authors gave the same response as above.)

Round 2

Reviewer 1 Report

The submitted revised article “sensors-431579-v2” entitled: “Damage quantification with embedded piezoelectric aggregates based on wavelet packet-based energy analysis” has substantially been improved and ameliorated. The efforts performed by the Authors to consider all the recommendations of the previous review are greatly appreciated. The paper can be accepted for publication without further re-review.

There are only a few minor suggestions:

- The fonts of the text in Figs. 10, 11 and 15 should be enlarged.

- Reference list format should be modified in order to follow the instructions of the Journal (https://www.mdpi.com/journal/sensors/instructions#references).

Reviewer 2 Report

The authors reflected or explained all the comments